# The Role of Neutrophils in Spondyloarthritis: A Journey across the Spectrum of Disease Manifestations

**DOI:** 10.3390/ijms24044108

**Published:** 2023-02-18

**Authors:** Lavinia Agra Coletto, Chiara Rizzo, Giuliana Guggino, Roberto Caporali, Stefano Alivernini, Maria Antonietta D’Agostino

**Affiliations:** 1Division of Rheumatology, Università Cattolica del Sacro Cuore, Policlinico Universitario Agostino Gemelli IRCSS, 00168 Rome, Italy; 2Department of Health Promotion, Mother and Child Care, Internal Medicine and Medical Specialties, Rheumatology Section, University of Palermo, 90127 Palermo, Italy; 3Division of Clinical Rheumatology, ASST Gaetano Pini-CTO Institute, 20122 Milan, Italy; 4Department of Clinical Sciences and Community Health, University of Milan, 20122 Milan, Italy

**Keywords:** spondyloarthritis, neutrophils, psoriasis, uveitis, enthesitis, arthritis, inflammatory bowel disease

## Abstract

Spondyloarthritis (SpA) contemplates the inflammatory involvement of the musculoskeletal system, gut, skin, and eyes, delineating heterogeneous diseases with a common pathogenetic background. In the framework of innate and adaptive immune disruption in SpA, neutrophils are arising, across different clinical domains, as pivotal cells crucial in orchestrating the pro-inflammatory response, both at systemic and tissue levels. It has been suggested they act as key players along multiple stages of disease trajectory fueling type 3 immunity, with a significant impact in the initiation and amplification of inflammation as well as in structural damage occurrence, typical of long-standing disease. The aim of our review is to focus on neutrophils’ role within the spectrum of SpA, dissecting their functions and abnormalities in each of the relevant disease domains to understand their rising appeal as potential biomarkers and therapeutic targets.

## 1. Introduction

The term Spondyloarthritis (SpA) includes a group of different diseases sharing a common pathogenetic background and similar clinical features such as Ankylosing Spondylitis (AS), Psoriatic Arthritis (PsA), inflammatory-bowel-disease-related arthritis (IBD-related arthritis), and Reactive Arthritis (ReA) [1].

Skeletal manifestations involve both spine and peripheral joints, with a prominent role played by enthesis within the so-called synovio–entheseal complex, which is the anatomical structure linking tendons or ligaments to bones [2]. Extra-articular features of SpA are mostly represented by skin disease, with psoriasis or psoriasiform lesions, gut inflammation, both clinical and subclinical, and eye involvement, characteristically evolving in uveitis [3]. From a pathogenetic point of view, aberrant innate and adaptive immune responses, triggered by several environmental factors in a genetically predisposed subject, determine the development of a strong pro-inflammatory milieu that drives the systemic inflammation retrieved in SpA [4]. Given the absence of detectable serum autoantibodies and the relevance of Human Leucocyte Antigen (HLA) and T cells in the pathogenesis, SpA, as a matter of classification, has touched in the past the fine line of the autoinflammatory diseases spectrum, where innate immunity leads the way [5]. In the context of recent advances, amidst the three immunity types which reshaped the categorization of immune response [6], disrupted type 3 immunity stood out as the most pervasive trait in SpA pathophysiology [7]. Physiologically, extracellular disturbances in barrier tissues, for instance, caused by the interaction with bacteria and fungi, initiate type 3 response, where interleukin (IL)-17 and IL-22 orchestrate cellular compartments to protect the body against the damage with the aim of restoring tissue homoeostasis [6]. However, when abnormal, they contribute to SpA pathogenesis by amplifying T helper 17 cells (Th17) response, hence recruiting and activating neutrophils at different disease sites (enthesis, joint, gut, eye, and skin) in a positive loop, further amplified by neutrophilic production of pro-inflammatory mediators, such as IL-23 [8]. Hereby neutrophils are emerging as pivotal cells in SpA development, being involved in several manifestations, especially in the very early phase of the disease (Figure 1).

In the figure, the typical accumulation of neutrophils at the tissue level in the form of a cryptic abscess, Munro’s microabscess, and hypopyon is depicted within the intestinal mucosa, the skin, and the eye, respectively. At the entheseal level, the influx of neutrophils and their consequent activation drive enthesitis development and cooperate with the occurrence of new bone formation. Interleukin (IL), Tumor Necrosis Factor (TNF), Granulocyte-Macrophage Colony Stimulating Factor (GM-CSF).

### Neutrophils Overview

Every day, 10^11^ neutrophils are produced by the human bone marrow, making them the most abundant circulating leucocyte population, known as an essential component of innate immunity [9]. Due to their distinctive short lifespan, they have been historically perceived as a homogeneous and transcriptionally inactive cell population made for quickly responding to microbial and sterile challenges covering defensive and inflammatory duties. In the bone marrow, the maturation process from granulocyte–monocyte progenitors to segmented neutrophils is conducted by the granulocyte colony-stimulating factor (G-CSF). Once mature, neutrophils are mobilized from the bone marrow reservoir, upregulating CXC chemokine receptor 2 (CXCR2) and Toll-like receptor 4, whose ligands are expressed by endothelial cells [10], and downregulating the integrin α4β1 (VLA4) and the CXCR4 which usually binds resident stromal cells via vascular cell adhesion molecule 1 (VCAM1) and chemokine stromal-derived factor-1/SDF-1 (CXCL12), respectively [11].

Outside the bone marrow, the network orchestrating neutrophil function expands, and among the key regulators, IL-23 and IL-17 play a significant role in their activation [12]. Neutrophils themselves are capable of producing IL-17, perpetuating positive feedback with Th17 cells [13].

Hence, the interaction with activated endothelium via selectin and adhesion molecules together with a gradient of chemoattractant mediates neutrophils migration, adhesion, and diapedesis into inflamed tissue where they exert their function of clearance of extracellular pathogens or cell debris by multiple mechanisms, including phagocytosis and the release of reactive oxygen species (ROS), granules, and neutrophils extracellular traps (NETs).

Such static archetype has been challenged when neutrophils proved to modulate at different levels of innate and adaptive immunity, hence not limiting their role in the initiation and amplification of the inflammatory response [14]. Indeed, neutrophils may behave as antigen-presenting cells, able to produce chemokines, cytokines, exosomes, and granules targeting endothelium, dendritic cells, complement, macrophages, and lymphocytes with the result of upregulating or downregulating their function [15]. They were shown to populate secondary lymphoid organs under both homeostatic and inflammatory conditions, colonize peri-marginal zone areas of the spleen, display a B-cell-helper function contributing to immunoglobulin diversification, and engage bi-directional interaction with various subsets of T cells [15,16].

The advent of new techniques, such as single-cell RNA-sequencing (scRNA-seq) and pseudo-time analysis, revealed that functional diversity was mirrored by phenotypical diversity, starting to shape the heterogeneous trajectory of neutrophilic phenotypes, capable of aging and plasticity, both in health and disease [17,18].

Deregulation of neutrophils causes damage, both when defective, by worsening sepsis consequences, or hyperactive, contributing to tissue damage in diverse conditions, comprising cancer and immuno-mediated diseases such as SpA [18]. A comprehensive knowledge of neutrophil biology, functions, and deregulation in SpA may contribute to a deeper understanding of disease onset with a consequent important impact on the discovery of new therapeutic targets in SpA. In this narrative review, considering the pleiotropic role of neutrophils, we aim to evaluate their behavior in SpA, dissecting their function and abnormalities in each of the relevant disease domains.

## 2. Neutrophils across SpA Manifestations

### 2.1. Articular Involvement

#### 2.1.1. Enthesis

SpA clinical picture is dominated by significant articular and periarticular involvement, and enthesitis is considered the hallmark feature. Specifically, enthesis seems to be the primary site of inflammation responsible for the systemic activation of the immune system, and neutrophils are deeply involved in the very early phase of disease, as they mediate tissue damage through ROS and protease production [19].

Resident cells release pro-inflammatory mediators, including G-CSF, IL-17, IL-8, and prostaglandins, to induce vasodilation of the transcortical feeding vessels and to favor the active recruitment and diapedesis of neutrophils in loco [20,21]. This specific crosstalk is exemplified by the evidence of stromal cells, obtained from enthesis specimens, that once stimulated with fungal adjuvants, release IL-8 and upregulate leukocytes adhesion molecules, such as VCAM-1 and Intercellular Adhesion Molecule-1 (ICAM-1), key contributors to neutrophils migration.

Notably, Tamassia N. and colleagues showed, for the first time, the presence of neutrophils in healthy spinal peri-entheseal bone. These enthesis-derived neutrophils stimulated in vitro were able to release a significant amount of IL-23, a key cytokine in SpA pathogenesis and enthesitis development [22].

The presence and activation of neutrophils, together with a marked increase in the alarmins S100A8 and S100A9, were elegantly showed by Stavre Z. in *SKG mice* withcurdlan-indiced enthesitis, a common use animal model for chronic arthritis [23,24]. These aforementioned alarmins constitute calprotectin, a biomarker of inflammation for both SpA and IBD, in close relationship with the development of tendinopathy, SpA structural damage, psoriasis, and gut inflammation [25,26,27]. S100A8 and A9 may also have an autocrine effect, stimulating their margination and adhesion to the vessel wall to translocate into inflamed tissue. Thus, alarmin production could sustain neutrophil influx to enthesis in the early phase of inflammation [28].

In addition, the proinflammatory cytokine burst, evidenced in myeloid cells during enthesitis, seems to be related to hyperreactive signaling via Signal Transducer and Activator of Transcription 1 (STAT1), and in the absence of the STAT1 counterregulatory protein A20, entheseal inflammation occurs spontaneously [29], strengthening the relevance of myeloid compartment in enthesitis pathophysiology.

Obtaining enthesis specimens to perform histological and functional assays is challenging [30]; however, a new minimally invasive technique to perform entheseal biopsy of the elbow common extensor tendon was recently described. The authors not only showed the ability to sample proper entheseal tissue but also demonstrated the presence of CD45^+^ cells in the digested tissue [31]. This finding paves the way to the possibility of reaching a comprehensive characterization of the entheseal cellular environment.

#### 2.1.2. Peripheral and Axial Joints

Together with enthesitis, SpA patients classically suffer from joint inflammation at both axial and peripheral levels. The prevalent involvement of the spine and sacroiliac joints defines the axial SpA (axSpA) phenotype; while in the presence of mainly oligoarthritis or polyarthritis and in the absence of axial involvement, the disease is classified as peripheral SpA (pSpA) [32].

Histological data reported the presence of neutrophils at both levels, pointing out a potential role in triggering the initial inflammatory cascade in SpA. In specimens from early sacroiliitis, neutrophils were evidenced among other immune cells, taking part in the enthesitis process in such areas [33]. In addition, in facet joints tissue of AS patients, a strikingly high number of IL-17^+^ neutrophils, as demonstrated by double immunofluorescence assays with both CD15 and Myeloperoxidase (MPO), was found, suggesting a role even in the advanced stage of disease and particularly in the occurrence of syndesmophytes and enthesophytes [34].

Sampling tissues from the axial skeleton is difficult, but neutrophils presence and activity in SpA spinal inflammation and consequent structural changes are inferable by the increased levels of Granulocyte-Macrophage Colony Stimulating Factor (GM-CSF) in patients with persistently active disease [35]. Indeed, enhanced granulopoiesis and neutrophil differentiation are evidenced in inflamed tissues of SpA patients, and they are mirrored peripherally in the blood by the increase in neutrophil count, findings confirmed in animal models of disease [36]. Neutrophil-to-leukocyte ratio, when elevated, has been proposed as a marker of disease activity, linking neutrophils to clinical manifestations of SpA and disease activity scores [37] and the increased number of granulocytes precursors, as a marker for structural damage in axSpA [38].

From a functional point of view, in the peripheral blood of patients affected by radiographic axSpA with active clinical disease, a spontaneous excessive Neutrophil Extracellular Traps (NETs) generation was depicted with an increase in NETs remnants, such as cell-free DNA, cell-free nucleosomes and elastase [35]. These NETotic products correlate with inflammatory markers and may be used to discriminate active patients [39]. Notably, NETs obtained from AS patients are enriched with bioactive IL-17A and IL-1β. IL-17A induces the differentiation of mesenchymal stem cells toward bone-producing cells, whereas IL-1β contributes, in a pro-inflammatory loop, to determine IL-17 accumulation in NETs. To reinforce this concept, the blockade of the IL-1 β pathway reduces the osteogenetic differentiation of mesenchymal stem cells [40].

Taken together, these lines of evidence may pinpoint an active role for neutrophils and their upstream stimulation in SpA initiation and progression, implying an unrecognized plasticity of these cells across SpA phases with specific involvement in spinal damage [41,42].

Peripheral SpA may account for a neutrophil role as well. The first description of neutrophils in synovial fluid from AS-inflamed knees dates back to 1973 [43]. Later, tissue analysis of synovial membrane from inflamed joints revealed the presence of neutrophil infiltrate. Polymorphonuclear leukocytes (PMNs) infiltrates are comparable among all SpA subsets (axial and peripheral), including PsA, and are increased when compared to rheumatoid arthritis [44].

In active SpA, PMNs infiltration showed a strong correlation with C-Reactive Protein (CRP) and Erythrocyte Sedimentation Rate (ESR), demonstrating a close relationship with systemic inflammation and clarifying that local inflammation mirrors global disease activity [45]. Moreover, the presence of abundant PMNs infiltrate in joints of all SpA subtypes in the initial phase of disease reinforces the concept that innate immunity may act as the primary driver in determining peripheral joint inflammation. On the other hand, the reduction of PMNs at the joint level is obtained after treatment, as demonstrated with TNF inhibitors in early SpA [46], making histological changes a useful biomarker to assess response to treatment.

Recently, a marker of NETosis, the MPO–DNA complex, was shown to be increased in PsA peripheral blood as compared to PsO, suggesting a stronger inflammatory burden in PsA and a more robust occurrence of NETosis in such patients. The MPO–DNA complex correlated with disease activity parameters in PsA, especially with tender and swollen joint counts, as well as with PsA disease activity scores (DAPSA, Disease Activity in PSoriatic arthritis), indicating that its serum fluctuation may serve as a marker to catch articular involvement, and even response to drugs [47]. NETs in the context of PsO were demonstrated to induce Th17 cells, with a stronger induction in the presence of TRAF3 Interacting Protein 2 (TRAF3IP2) genotype [48], renowned for increasing the susceptibility of developing PsA and associated with a more severe joint involvement in PsA [49]. We can hypothesize that, as observed in PsO, NETs may enhance Th17 response in PsA, contributing to orchestrating tissue and systemic inflammation via the IL-17 axis. Indeed Cathelicidin Antimicrobial Peptide LL 37 (LL37), a product of neutrophil degranulation, is highly represented in PsA synovium [50]. Neutrophil degranulation is boosted in the presence of the complement factor, C5a, and of GM-CFS, both increased in PsA synovial tissue. Once released, LL37 may elicit the generation of anti-LL37 autoantibodies that can contribute to shape the pro-inflammatory milieu in PsA favoring immune-complex (IC) deposition, that in turn fuels NETosis in a self-maintained loop. Indeed, neutrophils were shown to be IC^+^ synovial cells, presenting co-localization of IgG and LL37 on immunofluorescence examination, mainly in synovial tissue of PsA patients hosting lymphoid aggregates, so they could act as source of antigens and, simultaneously, as target of antibody-mediated immune response [51].

### 2.2. Gut Involvement

The link between gut inflammation and SpA emerged in the last decades following the clinical observation that up to 10% of SpA patients develop IBD, mainly as Crohn’s disease (CD), and more than 50% present subclinical gut inflammation at histological examination, without overt IBD manifestations [52]. The actual knowledge of neutrophil functions in gut inflammation is still incomplete. Evidence from IBD studies pointed out a double-edged behavior of these cells, as they can contribute to inflammatory changes and tissue damage and, simultaneously, to tissue healing and inflammation resolution [53]. In CD, crosstalk between Th17 and neutrophils was recently described, where the two populations of cells co-localized in the mucosa, suggesting a vicious bidirectional loop [13]. A similar mechanism, if present in SpA, could contribute to the IL-17-driven inflammation evidenced in AS and PsA.

The implication of neutrophils in driving gut inflammatory changes in SpA patients may be inferred by the close relationship with microbiota dysbiosis and the presence of an altered gut-vascular barrier (GVB), making easier the translocation of pathogens and their products to the lamina propria with the consequent triggering of immune responses [54,55]. Specifically, NETs were shown to further elicit bacterial translocation from the lumen and promote the apoptosis of epithelial gut cells, contributing to the disruption of intestinal barrier integrity [56].

The integrity of GVB relies on type 3 immunity with a delicate equilibrium of cytokine production to grant tight junction functions, mucus production, and, ultimately, gut homeostasis [57,58]. SpA patients exhibit a well-known unbalance in type 3 cytokine response, specifically IL-23, IL-17, and IL-22, which contributes to the disruption of gut epithelial integrity resulting in the occurrence of the so-called “leaky gut” [59]. Neutrophils are attracted by IL-8 and actively released by resident epithelial cells.

Once there, they can be considered both initiators of the inflammatory process and active contributors of mucosal damage by recognizing Pathogen Associated Molecular Patterns (PAMPs), acting as antigen-presenting cells (APCs), releasing toxic molecules (including reactive oxygen species (ROS), matrix metalloproteases (MMPs), and NETs), cytokines (such as CXCL8 and IL-17) and alarmins, ultimately boosting a strong pro-inflammatory cascade and shaping the inflammatory microenvironment [60,61,62]. The presence of neutrophils in the inflamed intestinal mucosa is corroborated by increased levels of calprotectin (S100A8 and S100A9) in SpA feces and serum [63] and by histopathological findings. Indeed, in SpA subclinical ileal inflammation, an acute form of colitis strikingly resembling bacterial enterocolitis was found, with massive influx of PMNs within the lamina propria and the epithelium [64]. Such picture is retrieved in both AS and PsA [65] and correlates with a predominant peripheral disease. On the other hand, chronic changes, accounting for a profound architectural distortion of the mucosal ultrastructure in the presence of immune infiltrate and aggregates in the lamina propria, are characteristically related to established axial disease [66,67].

However, neutrophils exert scavenger functions and may directly produce anti-inflammatory cytokines, such as IL-10 and resolvins, that mediate tissue remodeling and mucosal healing [68,69]. The counter-regulation of their activity is mainly determined through apoptosis and consequent reduction of neutrophil infiltrate [70,71].

Notably, the presence of an increased level of GM-CSF in IBD favors neutrophil survival, inhibiting apoptosis [72,73]. GM-CSF is upregulated even in SpA, and a common detrimental function on neutrophil homeostasis may then be present in this condition as well.

Neutrophils display tissue-specific and microenvironment-driven functions; they are short-lived cells recruited within minutes with a peak response occurring in 24–48 h, thus hard to study in action [74]. However, their effects go beyond such a time frame since they favor the local polarization of immune cells, which can recirculate from the intestinal niche to target sites of disease, including joints, enthesis, eyes, and skin [75].

### 2.3. Psoriasis

PsO is an inflammatory skin disease rather prevalent (1–5%) worldwide [76] and in approximately 10 to 30% is associated with or may precede PsA occurrence [77]. The crosstalk between innate and adaptive immune systems leads to a self-perpetuating inflammatory loop played by T cells (mainly Th1, Th17, and Th23 driven), dendritic cells, keratinocytes, and neutrophils [78].

In the skin, neutrophils are the earliest cells present in psoriatic plaques, and when clustering, they are responsible for pathognomonic histological hallmarks, such as Munro’s microabscesses and Kogoj spongiform micropustules [76].

In the blood of psoriatic patients, neutrophils are augmented in terms of absolute count [79], ratio (neutrophil-to-lymphocyte ratio) [80], and of subsets (low-density granulocytes (LDG) and normal-density granulocytes (NDG). Both the absolute count and circulating LDG are associated with psoriasis skin disease severity (PASI: Psoriasis Area Severity Index) [81]. Cytokines associated with neutrophil recruitment, differentiation, activation, and release are increased in the plasma of psoriasis patients compared to controls [79].

Conversely, therapeutic depletion of myeloid lineage leukocytes, whether voluntary or involuntary, is associated with a rapid improvement of psoriasis [82,83], and targeted biologic therapies for psoriasis simultaneously ameliorate clinical symptoms and normalized neutrophil activity and count [84].

Neutrophil alterations are observed at all stages, comprising respiratory burst, degranulation process, and NETosis [85].

They produce and release a larger amount of ROS compared to healthy individuals. The activity of NADPH oxidase (NOX2) and MPO, two key enzymes responsible for the respiratory burst, is increased both in the skin and serum of PsO patients and correlates with PASI severity [86,87]. Enzymes stored and then released by neutrophils granules are implicated into the pathogenesis of psoriasis: (i) proteinase 3 is involved in the proteolytical activation of inflammatory mediators (such as IL-36 and Tumor Necrosis Factor (TNFα)) [78,88] and in the formation of autoantigens in psoriasis (LL37) [89], contributing both to local and systemic inflammation; (ii) neutrophil elastase (NE) enhances keratinocyte proliferation via proteolytic activation of epidermal growth factor receptor (EGFR) signaling [90] and together with (iii) cathepsin G contribute to IL-36 activation and subsequent skin inflammation [88]; (iv) the relevance of MPO, already mentioned for its role in the respiratory burst, is also evinced by genetic and functional studies where the gene encoding for MPO resulted as a genetic determinant of generalized pustular psoriasis and neutrophil abundance [91]; (v) finally lipocalin 2 (LCN2) is an antimicrobial protein which seems to enhance Th17-mediated inflammation, and whose serum level is elevated in psoriatic patients and correlate with the severity of itching [92].

Finally, NETosis is more than a bystander effect of neutrophil dysregulation in psoriasis. NETs are increased both in psoriatic lesions [93,94] and blood samples of psoriatic patients, where they correlate with PASI [95,96]. They act as key players in inducing and maintaining inflammation in different manners: (i) DNA-LL37 complexes stimulate the production of Interferon-alpha (IFN-α) and IFN-β by plasmacytoid dendritic cells (pDCs) [97], whereas (ii) RNA-LL37 complexes stimulate TNF-α and IL-6 production by myeloid DCs [98,99], (iii) NETs signaling, through activation of Toll-Like Receptor-4 (TLR-4) and IL-36, amplifies skin inflammation [94] and (iv) IL-17A, a central cytokine in psoriasis pathogenesis, is released by neutrophils during the formation of NETs in psoriatic lesions [100].

To close the loop, gene ontology enrichment analysis of genes expressed in psoriatic lesions also revealed significantly upregulated genes involved in neutrophil modulation [101], reinforcing neutrophils’ relevance in the pathogenesis of psoriasis.

### 2.4. Uveitis

Uveitis is the most common extra-articular manifestation in SpA, occurring preferentially in *HLA-B27*-positive patients. Up to 50% of AS and only 7% of PsA, and 2–5% of IBD-SpA patients develop uveitis [102].

Uveitis can be classified as anterior, intermediate, or posterior, according to the portion of the eye involved in the inflammatory process. In the case of global inflammation, the disease is described as panuveitis. The direct detection of leukocytes in a fluid near the uveal tract or the visualization of choroid or retina lesions allows the diagnosis [103]. Uveitis displays differential clinical presentations across SpA: in AS, it is classically anterior, unilateral, recurrent, more frequent in males and with sudden onset; while in PsA, it is anterior and intermediate, bilateral, chronic, affecting more often females and with an insidious onset [104].

As evidenced in studies of patients with acute anterior uveitis (AAU) and animal models, innate immunity cells, including neutrophils, play a role in both SpA and ocular autoimmune disease [105].

In the early phase of the disease, neutrophils and macrophages massively infiltrate ocular structures with concurrent edema, increase in vascular permeability, and congestion. In acute uveitis, neutrophils secrete several proinflammatory cytokines, such as IL-1, IL-18, IL-36, and TNFα, as well as ROS and lysosomal enzymes [106]. Especially in *HLA-B27* subjects, they contribute to the accumulation of such proinflammatory cytokines in the aqueous humor taking part in non-granulomatous inflammation [107] and rarely, in the case of tissue necrosis, giving rise to real abscesses.

In acute uveitis and in case of relapsing disease, hyperreactive neutrophils may be responsible for tissue damage [108] and for triggering an inflammatory loop that can simultaneously affect the eye, joints, and skin. In detail, the pathophysiology of anterior uveitis generally involves the presence of chemoattractants in the corneum and the hyperreactivity of neutrophils via the increase in superoxide production and chemotaxis, as evidenced in models of AAU [109,110]. A nice report on a PsA patient demonstrated the link between peripheral neutrophil activation and eye disease [111]. In particular, iridocyclitis characterized by severe hypopyon (reflecting neutrophil infiltration of the anterior chamber) was coupled with the occurrence of arthritis and the worsening of psoriatic lesions. Moreover, the peripheral neutrophils of the patient were shown to be in an activated state during the disease flare and normalized after treatment.

An aberrant response to bacterial molecules in neutrophils has been proposed as a leading mechanism in the pathogenesis of AAU. In active AAU, including AAU related to SpA, there is a selective perturbation in the expression and function of TLR-2 and TLR-4. Such receptors are involved in recognition of bacterial PAMPs, namely lipoproteins and LPS, and can be internalized after the engagement with their ligands. In peripheral neutrophils obtained during AAU flare, a significant reduction in the levels of TLR-2 expression was observed, suggesting that neutrophils may have been activated by microbial products with consequent downregulation of surface receptors. Moreover, in vitro stimulation of patients’ neutrophils with TLR ligands resulted in an increased production of IL-1, especially after TLR-2 activation [112]. The observation that LPS derived from Chlamydia trachomatis (a pathogen strongly implicated in *HLA-B27* associated AAU and ReA) signals via TLR-2 strengthens the hypothesis of a possible alteration in neutrophils response to PAMPs and the occurrence of uveitis in the context of SpA [113,114].

The role of neutrophils in ocular inflammation is further evinced from studies on experimental autoimmune uveoretinitis. In particular, G-CSF and G-CSF-R are elevated in the ocular tissue and blood of patients with uveitis [115]. G-CSF may act both at the bone marrow level, determining an increased mobilization of neutrophils with consequent neutrophilia, and at the tissue level, enhancing sterile inflammation mediated by neutrophils. In this regard, ocular endothelial cells and Th-cells are accounted as a source of G-CSF, which leads to the release of neutrophils chemoattractants, such as CXCL2 and IL-8 [10]. Local G-CSF favors the expression of adhesion molecules in the endothelium itself and of chemokine receptors on neutrophils (CXCR2 and CXCR4), thus promoting their trafficking to the inflamed eye and, at the same time, boosting their survival [116]. Moreover, G-CSF-activated-neutrophils discharge proinflammatory cytokines, including IL-17 and IL-1, that influence Th17 polarization via effects on dendritic cells and IL-23 release [100,117], highlighting the inflammatory circle described in the pathogenetic process typical of SpA.

Intriguingly, the genetic deletion of G-CSF or the administration of monoclonal antibodies selectively blocking G-CSF caused a profound reduction in the severity of ocular manifestation in uveoretinitis models [118]. The decreased activation of neutrophils impaired even the differentiation of pathogenic Th17, bridging the two branches of immunity and showing up the fascinating hypothesis that neutrophils are more than ancillary cells during sterile inflammation, being able to actively contribute to uveitis development.

To conclude, a clue for neutrophil involvement in uveitis related to SpA can be inferred from anatomical studies, as McGonagle and colleagues proposed that the connective tissue of uveal structures, especially at the level of ciliary body tendons attachment, may be considered as a classical musculoskeletal enthesis. The same group, as mentioned above, recently described the presence of neutrophils at entheses so that we can speculate a possible common immunologic milieu for skeleton and eye structures, both contributing to SpA manifestations [24].

Taken together, these findings suggest neutrophils as key players in the systemic inflammatory process of SpA, intertwining different niches [111,119].

How neutrophils can be triggered in SpA to determine marked inflammation across multiple tissues is a matter of debate. In this regard, an intriguing unifying theory has emerged in the last few years accounting for a link between gut and eye inflammation, the so-called gut–eye axis [120]. This observation in SpA is coupled with the renowned hypothesis of the gut–joint axis in which the leaky gut may account for the spreading of bacterial antigens to several tissues, including the intraocular environment, as well as for the local activation of immune cells that can then recirculate to target sites of disease [121].

## 3. Therapeutics: Neutrophil-Targeted Therapies

Given the cross relevance of neutrophils among the most diverse pathologies with a huge epidemiological impact (such as sepsis, oncologic and autoimmune diseases), pharmacological research has implemented the development of drugs aimed at extinguishing, enhancing or modulating neutrophils count, function, or phenotype, when derailed [122].

We addressed the pathogenetic abnormalities concerning neutrophils across the various domain in the spectrum of SpA, and the main strategy to counterbalance the damage caused by neutrophils would be to switch off or attenuate their functionality, trying to contextualize them within the delicate cellular crosstalk.

This could be achieved by several approaches, including directly modulating neutrophils production, maturation, release into the circulation, migration, and accumulation in target inflamed tissues; by regulating their vitality or their mediators; and by indirectly modulating transduction signaling, albeit not specific only for neutrophils.

These are summarized in Table 1, with a special focus on drugs approved or attempted in clinical trials conducted on patients affected by autoimmune diseases.

Results are overall promising; however, some challenges, especially concerning safety and selectivity, remain. The major concern about safety is related to the risk of exacerbating severe infections, which is a recurrent challenge for rheumatologists when administering immunosuppressive therapies. Nonetheless, drug-induced neutropenia not always means interference with neutrophil-mediated antimicrobial host defense, as reported, for example, for IL-6 inhibitors and Janus Kinase (JAK) inhibitors [123,124], where neutropenia could serve as an indicator of efficacy without necessarily mirroring an increased rate of infections [125].

When targeting neutrophils, selectivity, and specificity represent the major technical challenge, which needs to be better addressed. Given the similarities and overlapping regulation of neutrophils with the other myeloid lineages, including macrophages, dendritic cells, monocytes, and osteoclasts, side effects might include disturbance of immunity and possibly bone metabolism.

Increasing the knowledge and characterization of neutrophil subsets and phenotypes [126] will help target specific neutrophil subpopulation, hopefully narrowing the desired effect on the pathological phenotype or dysfunction.

**Table 1 ijms-24-04108-t001:** Inhibition of neutrophils: therapeutic approaches in autoimmune diseases.

Strategies	Approaches	Target	Drugs and Diseases
Reducing neutrophil numbers	Targeting production	- GM-CSF receptor	*Mavrilimumab* in GCA and RA [127,128,129]
		- GM-CSF	*Otilimab* in RA [130]; *Namilumab* in RA [131], PsO [132] and SpA (NCT03622658); *Gimsilumab* in AS (NCT04205851; NCT04351243)
		- IL-23/IL-17 axis (a regulator of G-CSF production)	*IL-17 inhibitors* in PsO, PsA, ax-SpA; *IL-12/23 inhibitors* in PsO, PsA [133], CD, and SLE; *IL-23 inhibitors* in CD, UC, PsO, and PsA
	Inducing depletion	- Circulating neutrophils	-*Extracorporeal granulocytapheresis* in RA [134], CD, and RCU [135]
Interfering with neutrophil recruitment and chemotaxis	Selectin and integrin blockers	- α4β1-integrin	*-Natalizumab* in CD [136]
		- Selectins	-TBC1269 (and others) in PsO [137]
	Blocking complement	- C5a and C5a receptor	*-Eculizumab* and *Avacopan* in AAV [138,139]; *NNC0215-0384* in RA (NCT01611688)
	Blocking leucotriens	- LTB4	-*CP-195543* in RA (NCT00424294)
Blocking neutrophils activation	Signal transduction blockade (cytokine signaling in neutrophils)	- JAK	-*Jak-inhibitors* in PsA, RA, UC, AS [140]
		- SYK	-*Fostamatinib* in RA [141] and SLE [142]
		- PDE4	-*Apremilast* in PsO, PsA [143], SLE (NCT00708916) and AS [144]
	Blocking cytokines whose receptors are also on neutrophils	- TNF-α	-*TNF-α inhibitors* in RA, PsA, PsO, SpA, AS, CD, UC, uveitis [145]
		- IL-6	-*IL-6 inhibitors* in RA, AS, SSc, vasculitis, SLE, AOSD [146]
Blocking neutrophil-derived mediators	Neutrophil granule enzymes	- MMP9	-*andecaliximab* in UC and CD [147,148]
	NETs (blocking activity of enzymes critical for NETs formation)	- NADPH, MPO, PAD4, DNase Inhibitor	Not yet elucidated in humans affected by autoimmune diseases.
Others	Blocking neutrophil function	- Neutrophil inflammasome	-*β-hydroxybutyrate* in gout flares [149] -*IL-1β inhibitors* in RA, SpA, PsA, AS, AOSD, uveitis, GCA, vasculitis [150]
		- Neutrophils alarmins (S100A8/S100A9)	-*Paquinimod* in SLE (NCT00997100)

Abbreviations: Granulocyte-Macrophage Colony Stimulating Factor (GM-CSF), Interleukin (IL), (LTB4), (JAK), (SYK), Tumor Necrosis Factor (TNF), phosphodiesterase-4 (PDE4), Metalloproteinase (MMPS), Nicotinamide adenine dinucleotide phosphate (NADPH), ANCA-Associated Vasculitis (AAV), Myeloperoxidase (MPO), Rheumatoid Arthritis (RA), Psoriatic Arthritis (PsA), Psoriasis (PsO), Spondyloarthritis (SpA), Ankylosing Spondylitis (AS), Ulcerative Colitis (UC), Crohn Disease (CD), Giant Cell Arteritis (GCA), Adult Onset Still’s Disease (AOSD), Systemic Lupus Erythematosus (SLE).

## 4. Conclusions

The definition of SpA itself is a tough task due to the extreme heterogeneity of clinical manifestations, where apparently distant organ involvements are each defined by one specific disease. Yet, joints, spine, gut, skin, and eyes diseases are often clinically associated, embracing one another and sharing a common pathogenic background, driven by type 3 immunity, as recently demonstrated by Nakamura et al. [151].

In this complex and delicate picture, neutrophils designate a common thread, being a cell cross-sectionally involved in the pathogenesis within each of the addressed domains.

They indeed represented the bulky inevitable cell, depicting pathognomonic lesions, faithfully responding to the acute inflammatory duties. Their peripheral count and biological functions were shown to be affected by multilayer evidence both at early and late stages of the disease, often mirroring disease activity.

Moreover, their pathogenic relevance increased, considering their multifaced role, not only as soldiers of the innate immune system but also capable of regulating homeostatic functions and engaging a bidirectional dialogue with adaptive immunity [152]. Neutrophils were shown to be capable of releasing a larger amount of IL-17A and of acting as APCs for T cells into already inflamed tissues [153,154]. Taken together, such evidence, coupled with the reviewed data, stress neutrophils once again as pivotal in SpA pathogenesis, making them attractive therapeutic targets, with the opportunity of exploiting diverse strategies and approaching all stages of their regulation.

Nonetheless, emergent technologies, both in the outpatient setting and the wet lab, are paving the way for deepening the knowledge on neutrophils’ pathophysiology and across SpA disease course, getting closer to the definition of novel biomarkers and treatment options.

## Figures and Tables

**Figure 1 ijms-24-04108-f001:**
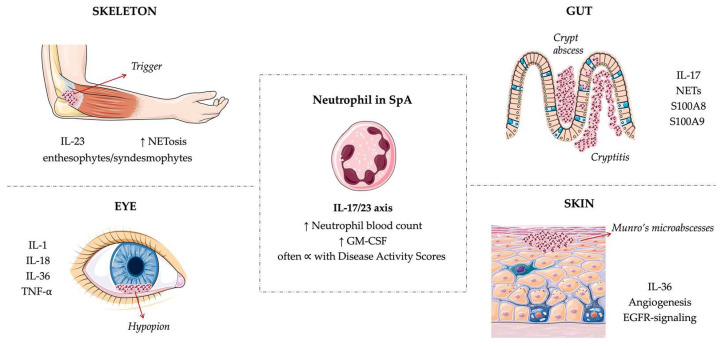
Legend. Neutrophils are pleiotropic cells that play a central role in the very early phase of SpA development, contributing to triggering inflammation in multiple tissues. Neutrophils are recruited from the bloodstream to target sites in Spondyloarthritis (SpA), such as joints, entheses, gut, skin, and eyes, where they produce neutrophils extracellular traps (NETs), cytokines, and chemokines to attract and activate other immune cells, boosting type 3 immunity.

## Data Availability

Not applicable.

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
