# Peer review of "The Role of Neutrophils in Spondyloarthritis: A Journey across the Spectrum of Disease Manifestations"

_ijms, 2023, doi:10.3390/ijms24044108_

Round 1

Reviewer 1 Report

The comprehensive review is very relevant and important for a wide readership.

The general scheme does not seem quite logical.

Thus, Spondyloarthritis (SpA) overview is in the sections 1. Introduction and 1.2 with description of neutrophils among them (part 1.1). It'd be reasonable to avoid unnecessary repetitions.

Unfortunatelly, polarization of innate immunity was not even mentioned.  Type 3 immunity is mediated by retinoic acid-related orphan receptor γt(+) ILC3s, TC17 cells, and TH17 cells secreting IL-17, IL-22, or both, which induce mononuclear phagocytes and neutrophils, thus protecting against extracellular bacteria and fungi with risk of development of autoimmune diseases.  Without such short description of main types of immunity, the simple list of key cytokines for immune response of type 3 (Th17 way) does not make sense.

Early stages and  "inflammatory stimuli" that "promote neutrophils mobilization from the bone marrow reservoir into the blood, followed by migration, adhesion, and diapedesis into inflamed tissues" are essential for molecular mechanisms of pathogenesis and should be considered in more detail.

Specific comments.

Without the abbreviation list at the beginning there are multiple repeats of some acronyms (for example, SpA) throughout the text. Similar acronyms hamper both reading and understanding.

There are the only Figure 1 and a single table that are evidently not enough to summarize all numerous publications (148 references).

Line 210. Gene expression combines RNA transcription and protein translation. The term "expression" is widely used for genes but not for peptides.

Minor problems are highlighted in yellow in attached file.

Hopefully, that the proposed changes will improve the manuscript and will be helpful for readers.

Author Response

We thank the Reviewer for his suggestions and here we provide a point by point reply to his comments:

-“The general scheme does not seem quite logical. Thus, Spondyloarthritis (SpA) overview is in the sections 1. Introduction and 1.2 with description of neutrophils among them (part 1.1). It’d be reasonable to avoid unnecessary repetitions.

-As suggested, we revised the scheme of the introduction eliminating the paragraphs 1.2 and summarizing the key concepts of SpA clinical characteristics in the “enthesis” (lines 143-147) and “peripheral and axial joints” (lines 267-271) sections in order to avoid unnecessary repetitions.

-“Unfortunatelly, polarization of innate immunity was not even mentioned.  Type 3 immunity is mediated by retinoic acid-related orphan receptor γt(+) ILC3s, TC17 cells, and TH17 cells secreting IL-17, IL-22, or both, which induce mononuclear phagocytes and neutrophils, thus protecting against extracellular bacteria and fungi with risk of development of autoimmune diseases.  Without such short description of main types of immunity, the simple list of key cytokines for immune response of type 3 (Th17 way) does not make sense.”

-As suggested, we better described the role of type 3 immunity in SpA (line 50-62), stressing its role in protecting against bacterial and fungal insults as well as in inducing and sustaining SpA development when disrupted.

-“Early stages and  “inflammatory stimuli”; that “promote neutrophils mobilization from the bone marrow reservoir into the blood, followed by migration, adhesion, and diapedesis into inflamed tissues”; are essential for molecular mechanisms of pathogenesis and should be considered in more detail.”

-As suggested, we expanded the section regarding neutrophils mobilization from the bone marrow and consequent migration, adhesion, and diapedesis into inflamed tissues, deeply describing the main molecules and pathways involved in all the above mentioned processes (lines 87-104).

Specific comments.

-“Without the abbreviation list at the beginning there are multiple repeats of some acronyms (for example, SpA) throughout the text. Similar acronyms hamper both reading and understanding.”

-We added a list of abbreviation at the beginning of our manuscript, as suggested (lines 29-33).

-“There are the only Figure 1 and a single table that are evidently not enough to summarize all numerous publications (148 references).”

-We summarized the main functions of neutrophils across the wide spectrum of SpA manifestation in figure 1, trying to keep it understandable, considering the complex role of neutrophils in SpA pathogenesis.

In table 1 we summarized all clinical trial in humans designed to test drugs or molecules targeting neutrophils functions in different autoimmune disease to give a perspective of the possible therapeutic approaches that can be developed in the near future.

A table summarizing all the cited paper would be very long and difficult to read, so we preferred to focus our attention on possible treatments. However, all relevant paper describing neutrophils in SpA pathogenesis are cited and commented in the main text.

-“Line 210. Gene expression combines RNA transcription and protein translation. The term “expression”; is widely used for genes but not for peptides.”

-As suggested, we changed “expressed” with “represented” (line 338)

-“Minor problems are highlighted in yellow in attached file.”

-As suggested, all the highlighted items were changed to make each sentence clearer.

Hopefully, that the proposed changes will improve the manuscript and will be helpful for readers.

Reviewer 2 Report

In Table 1- Author should consider citing literature targeting Neutrophil NETosis function in autoimmune diseases like Rheumatoid arthritis.

Author Response

We thank the Reviewer for his suggestion.

In table 1 we cited all relevant drugs or molecules targeting neutrophils in autoimmune diseases that were tested in clinical trials in humans. To the best of our knowledge, we did not find any registered trial on NETs inhibition in rheumatoid arthritis. However, other therapeutic approaches attempted in rheumatoid arthritis are reported in table 1.

Reviewer 3 Report

Authors Coletto et.al, neutrophils role in the spectrum of Spondyloarthritis Comments: 1. Authors consider citing  a recent paper from Macleod et.al, 2023
(https://doi.org/10.1016/S2665-9913(22)00334-4)

2. stavre et.al ( https://doi.org/10.1186/s13075-021-02693-7) reported neutrophils with inducible IL‑23 production are present in uninflamed human entheseal sites, andneutrophils are prominent in early murine spondyloarthritis‑related enthesitis.
, Authors consider citing this paper.

Author Response

We thank the Reviewer for his comments.

As suggested, we cited the very recent paper from Macleod et al., as you can find in line 60 (reference 8).

The article by Stavre et al. was already commented in the “enthesis” section, as you can find in line 161 (reference 24).

Round 2

Reviewer 1 Report

The revised manuscript was essentially improved. I believe that the only scheme of Figure 1 represents  rather Graphycal Abstract of the whole review. Additional illustrations would be helpful for readers. But I completely rely on your choice.

No additional comments are available.

Author Response

We thank the Reviewer for his precious suggestions. We took into consideration creating other illustrations, but ultimately we agreed they would further lengthen the paper, already dense.

Reviewer 2 Report

Accept in present form.

Author Response

We thank the Reviewer for his precious suggestions